# Immunofluorescent Localization of Plakoglobin Is Altered in Endomyocardial Biopsy Samples from Dogs with Clinically Relevant Arrhythmogenic Right Ventricular Cardiomyopathy (ARVC)

**DOI:** 10.3390/vetsci8110248

**Published:** 2021-10-23

**Authors:** Eva M. Oxford, Romain Pariaut, Massimiliano Tursi, Philip R. Fox, Roberto A. Santilli

**Affiliations:** 1Department of Clinical Sciences, Cornell University College of Veterinary Medicine, Ithaca, NY 14853, USA; rp223@cornell.edu (R.P.); rs2259@cornell.edu (R.A.S.); 2The Heart Vet, PLLC, Dryden, NY 13053, USA; 3Department of Veterinary Sciences, University of Turin, 10095 Turin, Italy; massimiliano.tursi@unito.it; 4Caspary Institute, The Animal Medical Center, New York, NY 10065, USA; philip.fox@amcny.org; 5Clinica Veterinaria Malpensa, Viale Marconi 27, Samarate, 21017 Varese, Italy

**Keywords:** boxer, English bulldog, intercalated disc, arrhythmias

## Abstract

Diagnosing the early stages of canine Arrhythmogenic Right Ventricular Cardiomyopathy (ARVC) is complicated by day-to-day arrhythmia variability, and absence of reliable, transthoracic echocardiographic features. Definitive diagnosis requires histopathologic identification of transmural fibrofatty replacement of the right ventricle. Reduction of immunofluorescent signal for plakoglobin (PG) at the intercalated disc (ID) is reported in ARVC-affected humans and boxers. Our objective was to determine whether reduced immunofluorescent signal for PG in endomyocardial biopsy samples (EMBs) correspond with a histopathologic diagnosis of ARVC. Here, 49 dogs were evaluated: 43 with advanced cardiac disease and 6 non-clinical boxers with mild to moderate ventricular arrhythmia (VA) burden. EMBs were obtained from all dogs; samples were prepared with antibodies recognizing cadherin (PC) and PG and evaluated with confocal microscopy. Investigators were blinded to breed and clinical status. ARVC was histopathologically diagnosed in 8 out of 49 dogs. Of these, three out of eight had clinical signs consistent with ARVC (two boxers, one English bulldog) and reduced PG signal at ID; five out of eight were non-clinical boxers with moderate VA and no reduction in PG. A total of 41 out of 49 dogs were histopathologically diagnosed with non-ARVC cardiac disease; 1 out of 41 showed reduction of PG at ID, while 40 out of 41 had no PG reduction. These results suggest that EMB PG signal is reduced in dogs with advanced ARVC, but not in the occult phase of the disease. Additionally, presence of PG at ID supports a diagnosis of non-ARVC cardiac disease in dogs with clinical signs. These results may offer an additional test that helps differentiate advanced ARVC from other myocardial diseases.

## 1. Introduction

Arrhythmogenic right ventricular cardiomyopathy (ARVC) is an important cardiac disease of the boxer [1] and English bulldog [2,3,4], resulting in dangerous ventricular arrhythmias, biventricular heart failure, and sudden cardiac death [1,4]. Histopathologically, this disease is characterized by myocyte replacement by adipocytes or by adipocytes and fibrosis (fatty or fibrofatty infiltration). While commonly first noted within the right ventricle, lesions may progress to involve the interventricular septum, left ventricle, and both atria [1,5,6,7]. Early clinical detection of ARVC is difficult, as structural cardiac changes are difficult to detect from transthoracic echocardiography. Furthermore, marked variability in day-to-day arrhythmia load is reported, [8] suggesting that 24-h Holter monitoring may have low sensitivity for diagnosis. Mutations in the striatin gene may be associated with some cases of ARVC in the boxer, [9,10] however, a definitive mutation in a larger subset of affected dogs remains unknown. While the presence of anti-desmoglein 2 (DSG2) antibodies have been recently found to be associated with ARVC in a small number of boxers, [11] wider studies are necessary to better understand the importance of this finding. As such, post-mortem histopathology of cardiac tissue is presently considered to be the only definitive method to diagnose ARVC [6,12,13,14].

The epidemiology and clinical profiles for ARVC are well characterized in the human population. Clinical and histopathologic characteristics are similar to those reported in the boxer and English bulldog [5,15,16]. As in the boxer, ARVC is difficult to definitively diagnose in humans in the absence of post-mortem histopathology. This has led to consideration of major and minor diagnostic criteria for humans, [12,14] although reliable criteria have not yet been established for the dog. In addition to the presence of anti-DSG2 antibodies in human subjects with ARVC [11], reduction of the immunofluorescent signal for the desmosomal protein plakoglobin (PG) in endomyocardial biopsy (EMB) samples is another potential factor to help diagnose ARVC [17,18]. Reduction of PG in ARVC affected cardiac tissue has been reported to occur throughout the right ventricle (RV), interventricular septum (IVS), and left ventricle (LV), even in the absence of fibro-fatty infiltration [16,19]. Therefore, detecting the loss of PG signal on human EMB samples improves the sensitivity of the EMB to detect ARVC [17]. Interestingly, loss of PG signal has also been reported in full thickness sections of myocardium from ARVC affected boxers [6].

Here, we hypothesized that reduced PG immunofluorescent signal in EMBs from boxers and English bulldogs would help to confirm a diagnosis of ARVC. We aimed to determine whether (1) reduction of PG in EMB could help to diagnose ARVC in dogs with clinical signs of cardiac disease, and (2) whether loss of PG in EMB could be used to differentiate ARVC from non-ARVC cardiac disease. Furthermore, we hypothesized that PG signal would not be reduced in dogs diagnosed with other cardiac disease.

## 2. Materials and Methods

### 2.1. Animals

This study took place in two parts. First, endomyocardial biopsies (EMBs) were obtained with owner consent from 43 client-owned dogs at the Clinica Veterinaria Malpensa (CVM) (Italy), as part of an unrelated study designed to identify etiologic agents for advanced myocardial disease. Excess tissue samples were used for this study. Clinical and histopathologic diagnoses for these dogs are included in Appendix A. For the second part of this study, EMBs were obtained from six clinically healthy boxers (>5 years), with mild to moderate ventricular arrhythmia (VA) burden diagnosed at Cornell University (CU) in accordance with an approved Institute for Animal Care and Use (IACUC) protocol (2016–0005). Prior to the procedure for the boxers, physical examinations, transthoracic echocardiograms, and 24-h Holter monitoring were performed (Appendix A). All dogs were free of clinical signs, had echocardiographic parameters within reference range, and had mild to moderate ventricular ectopic load (<10–5000 VPCs, no VT, see Appendix A) detected from 24-h Holter recording. These dogs were also suspected to have occult ARVC based on detailed pedigrees and previous research performed on this line of dogs [6]. Results for the striatin mutation for these dogs was either negative (via direct test) or unknown (one parent negative, one parent heterozygous; Appendix A). For this study, a clinical diagnosis of ARVC was made for dogs that met the following criteria: (1) breed (boxer or English bulldog), (2) documented ventricular tachycardia with a left bundle branch block morphology, (3) >100 ventricular premature complexes (VPCs) on 24-h Holter recording, and (4) fibrofatty infiltration of myocardium on EMB histopathology. Based on these criteria, dogs were divided into three groups: non-ARVC cardiac disease, clinical ARVC, and occult (non-clinical) ARVC.

### 2.2. Endomyocardial Biopsies

All dogs were anesthetized and 3 mm [3] samples of endocardium (3–6 separate samples from each dog) were obtained from the RV apical and interventricular septum region, using a bioptome introduced through the right jugular vein. Position of the bioptome was determined using fluoroscopic guidance, and the procedure was performed as previously reported [20]. The procedures were performed without immediate cardiovascular complications. However, one boxer (CU) presented to the emergency service three days after the procedure and was euthanized due to gastric dilation with volvulus (GDV). The anesthesia and stress of hospitalization were considered to be possible contributing factors in the development of GDV. The heart of this dog was collected after euthanasia, and a gross and histopathologic examination was performed by a collaborator (PRF) blinded to the signalment of the dog.

### 2.3. Sample Preparation

EMB samples were fixed for 48–72 h in 10% formalin and processed for paraffin embedding. Sections were treated with haematoxylin/eosin and Masson’s trichrome stain. Separately, sections were processed for immunofluorescence, as previously described [6,16]. Tissue sections were cut to 4 μm thickness, deparaffinized, and subjected to microwave antigen retrieval. Sections were then blocked at room temperature using 3% bovine serum albumin (BSA), 0.01% Triton buffer for 2 h, and incubated overnight at 4 °C with the appropriate primary antibodies. After incubation, samples were rinsed in phosphate buffered saline (PBS) and then incubated with the appropriate secondary antibodies at room temperature for an additional period of 45 min and mounted overnight before examination. Primary antibodies recognized pan-cadherin ((PC) product #C 3678, Sigma Adrich, Saint Louis, MO, USA) and plakoglobin ((PG) product #P 8087, Sigma Aldrich, Saint Louis, MO, USA). Hoescht dye (product #H3570, Thermofisher Scientific, Waltham, MA, USA) was used as a nuclear marker. The use of PC as an internal positive control has been previously described as a useful tool in confirming that tissue samples are properly fixed prior to treatment with antibodies [6,15,16]. Secondary antibodies used were Alexafluor 598 goat anti-mouse (Molecular Probes, Thermofisher Scientific, Waltham, MA, USA), and Alexafluor 484 goat anti-rabbit (Molecular Probes, Thermofisher Scientific, Waltham, MA, USA).

The whole hearts of the euthanized boxer, as well as from an English Bulldog with ARVC from the first part of this study (CVM), were fixed in 10% paraformaldehyde (PFA) and processed for gross pathology and histology with Masson’s Trichrome stain.

### 2.4. Confocal Microscopy

Confocal microscopy was performed using a Zeiss AxioObserver LSM 510 confocal microscope (Zeiss, White Plains, NY, USA) equipped with a 63X water immersion objective. The investigator (EMO) was blinded to the breed, clinical signs, and pathologic diagnosis of all samples obtained at CVM. For the second part of this study, performed at CU, the breed and clinical records of the boxers were known, however, the investigator (EMO) was blinded to the histopathologic diagnosis.

Diagnostic value of the EMBs was determined by the presence of PC, which indicated the location of intercalated discs (ID) on the samples. Conversely, the absence of PC signal inhibited further evaluation of the samples to establish a diagnosis of ARVC via immunofluorescence [6,15,16]. Samples were considered to have normal localization of PG when PG was colocalized with PC in the area of the intercalated disc (PG-ID), and mislocalization of PG (PG-MIS) when there was an absence of PG in the area of the ID. In PG-MIS samples, a perinuclear distribution of PG signal, as previously described in ARVC affected boxers [21], was often observed. Results were then compared to separately obtained histopathologic diagnoses.

### 2.5. Histopathology

A qualitative histopathologic diagnosis of ARVC was made by a pathologist blinded to the dog signalment (MT), based upon the presence of marked adipose or fibro-adipose infiltration of the EMB. (Appendix A). Adipocytes were of varied size and the fibrous component was identified as interstitial or replacement fibrosis. Moreover, additional findings associated with contiguous cardiomyocytes were considered, such as the presence of hypertrophic fibers with dysmetric or dysmorphic nuclei, and cardiomyocytes with sarcoplasmic vacuoles of varying sizes, indicative of a degenerative process. These findings allowed for ARVC to be distinguished from other cardiac diseases.

## 3. Results

ARVC was diagnosed by histopathology from EMBs in 8 of the 49 dogs, collected from both Clinica Veterinaria Malpensa (CVM) and CU. Of these, three were treated at CVM and had clinical signs consistent with ARVC (two boxers, one English bulldog). Five were non-clinical boxers from CU with a mild to moderate VA burden, and a family pedigree supporting an inherited link to ARVC. The remaining 41 dogs in this study were diagnosed with non-ARVC cardiac disease via EMB histopathology (non-ARVC cardiomyopathy (8); primary rhythm disturbance (7), myocarditis (9), viral heart disease diagnosed via positive polymerase chain reaction (PCR) assay (15), ischemic cardiomyopathy (1), no detectable disease (1)).

Figure 1A,B shows representative images from hematoxylin and eosin (H&E) stained samples from one dog diagnosed with lymphoplasmacytic myocarditis secondary to viral infection (Dog #29), and Masson’s trichrome and H&E stains revealing replacement fibrosis and infiltration of adipocytes in a dog diagnosed with ARVC (Figure 1C,D; Dog #49).

Within the subset of eight EMB samples with a histopathologic diagnosis of ARVC, PG signal was reduced at the ID and instead mislocalized in a perinuclear pattern (PG-MIS) in the three samples from dogs with clinical signs of ARVC (2twoboxers (Dogs #11, #21 Appendix A), one English bulldog (Dog #43 Appendix A)). Interestingly, PG-ID signal was present in all non-clinical boxers (Dogs #44–49 Appendix A), despite a histopathologic diagnosis of ARVC in five out of six. One boxer (#44, Appendix A) was euthanized due to GDV, and histopathology of full thickness sections of heart confirmed a diagnosis ARVC, despite the presence of PG-ID signal on EMB. The sixth non-clinical boxer (#47, Appendix A) was not diagnosed with ARVC via EMB, though mild adipose infiltrates were present. Plakoglobin signal was present at the ID in this dog (#47 Appendix A), along with the other non-clinical ARVC boxers.

Within the 41 EMB samples with a histopathologic diagnosis of non-ARVC cardiac disease, 5 samples were negative for PC signal and were not analyzed for PG signal. These samples were of small size, displayed improper tissue orientation, or had marked cellular infiltrates with few to no cardiomyocytes present. Of the 36 remaining EMB samples with non-ARVC cardiac disease, only 1 sample was negative for PG. This sample was from a mongrel with a presumptive diagnosis of myocarditis and additionally had fibro-fatty infiltration on histopathology. Figure 2 shows representative confocal microscopic images taken from four different dogs: a Dogue de Bordeaux with viral myocarditis and no reduced PG signal at the ID (Dog #25), an English bulldog with ARVC and loss of PG-ID signal and PG-MIS characteristic to ARVC [21], a non-diagnostic sample from a Great Dane with tachycardia induced cardiomyopathy, and a non-clinical boxer with a histopathologic diagnosis of ARVC and PG signal colocalized with PG-ID.

## 4. Discussion

In this study, we aimed to determine whether altered immunofluorescent PG localization in EMB samples could aid in the diagnosis of ARVC in boxer and English bulldogs with and without clinical signs. Surprisingly, while we hypothesized that PG-ID signal would be absent in all dogs with a histopathological diagnosis of ARVC, this hypothesis proved incorrect. Instead, we found that PG signal was present at the ID in EMB samples from all non-clinical boxers with a histopathological diagnosis of ARVC. Interestingly, PG signal was mislocalized in dogs with both clinical signs and a histopathological diagnosis of ARVC.

Notably, reduced PG-ID appeared to be highly specific in differentiating ARVC from other forms of cardiac disease in dogs with clinically relevant cardiac disease, as only 1 out of 36 diagnostic samples from dogs with non-ARVC related cardiac disease revealed reduced PG signal. More precisely, PG-ID signal was detected in 35 out of 36 EMB samples from cases of non-ARVC cardiac disease, while PG-ID signal was reduced in 3 out of 3 EMB samples from clinical ARVC cases. Interestingly, PG-ID signal was present in five out of five cases of non-clinical ARVC. This finding was surprising, as previous studies have shown that mislocalization of PG occurs in ARVC boxers [6], with similar results reported in humans [15,16,17]. It should perhaps be considered that ARVC-affected tissue samples are typically not studied in dogs lacking clinical signs. Rather, cardiac tissue samples are often obtained from those dogs with more advanced disease that are euthanized or died suddenly due to complications of ARVC [6], or when clinical signs justified the use of invasive testing, such as EMBs [4].

The composition of the intercalated disc, including localization of PG, has not yet been evaluated in the early stages of ARVC in dogs. There is much evidence that disruption of the intercalated disc is associated with the development of conduction disturbances, via loss of gap junctions [6,16,19,22], or ultrastructural defects [13]. It has been hypothesized that progressive disruption of the intercalated disc is correlated with the development of severe arrhythmias and clinical signs [6,13,15,23]. Hence, in the early stages of the disease, the intercalated disc may remain intact with proper PG localization or may become increasingly disrupted concurrent to progressive fibro-fatty infiltration of the myocardium. As such, these factors may play a role in the ability of diagnosing ARVC via EMB and thus, diagnosis of ARVC via EMB may remain dependent on the degree of disease progression. These potential relationships have not been clarified in the dog.

We evaluated six boxers with no reported clinical signs indicative of ARVC, unremarkable echocardiograms, and mild to moderate ventricular ectopic load on 24-h Holter recordings. These boxers are part of a pedigree with documented ARVC in the line via postmortem histopathology [6]. Based on these data, we suspected these dogs to have occult ARVC. Endomyocardial biopsy samples were all positive for PC and PG-ID signal, though five out of six of these boxers were positively diagnosed with ARVC via endomyocardial biopsy.

The presence of fibro-fatty infiltrates in the RV can be patchy, particularly in the early stages of the disease [15,17], and diagnosing ARVC on EMB is dependent on acquiring a number of samples and the sample location [24]. The single boxer not diagnosed with ARVC on EMB (#47) may have been unaffected, however, mild patchy infiltrates indicative of ARVC may not have been identified on EMB.

Mislocalization of ID proteins has not yet been described in English bulldogs affected with ARVC. However, it is suspected that the pathophysiology of the disease is similar to that in the boxer, as historically the boxer breed originated from the English bulldog [25]. Therefore, it is likely that the boxer and English bulldog may share a common molecular pathway leading to the development of ARVC. Here, we successfully identified an English bulldog, based on the absence of immunodetectable PG in a blinded study, that had been histopathologically diagnosed with ARVC, further supporting similarities between ARVC in the boxer and English bulldog.

Interestingly, our assay detected loss of PG-ID signal in a mongrel dog (Dog #22) diagnosed on histopathology with acute myocarditis. Three different viruses were detected in EMB samples from this mongrel via PCR (Adenovirus 1, Parvovirus, enteric Coronavirus), leading to the diagnosis of viral myocarditis. This EMB tissue sample also contained notable fat and fibrofatty replacement, the former of which is not a typical finding in acute myocarditis. It should be noted that inflammatory infiltrates have been reported in severe cases of ARVC in humans, and hypothesized to modulate the severity of disease [1,11,18,26]. Post-mortem histopathology was not performed on this dog, therefore a definitive diagnosis was not able to be confirmed. Furthermore, ARVC has been reported previously in breeds without a known familial link [27,28,29,30,31,32]. Whether the loss of PG signal in this case was a false positive result, or that this dog (a mongrel with unknown heritage) may have been affected by ARVC remains unknown.

Our data set are limited in that of the 49 EMB samples considered of diagnostic quality, only a small subset had a histopathologic diagnosis of ARVC (three dogs with clinical signs of severe cardiac disease, and five boxers with no clinical signs). Therefore, positive and negative predictive values may not be precise indicators of diagnostic accuracy [33].

Reduction of PG-ID in EMBs corresponded well to those dogs showing clinical signs of cardiac disease and histopathological evidence of ARVC. Furthermore, EMB histopathology was quite effective in detecting cellular infiltrates indicative of ARVC in dogs with and without clinical signs. Finally, the presence of PG-ID was highly correlated to non-ARVC cardiac disease in dogs with clinical signs. Therefore, we suggest concurrent that testing of localization of PG via immunofluorescence assays may be useful in conjunction to traditional histopathology in dogs with clinical signs undergoing an EMB procedure. Limitations notwithstanding, reduction of PG signal may offer additional evidence of ARVC in those cases of patchy RV fibro-fatty infiltrate that are not detected in small EMB samples.

## Figures and Tables

**Figure 1 vetsci-08-00248-f001:**
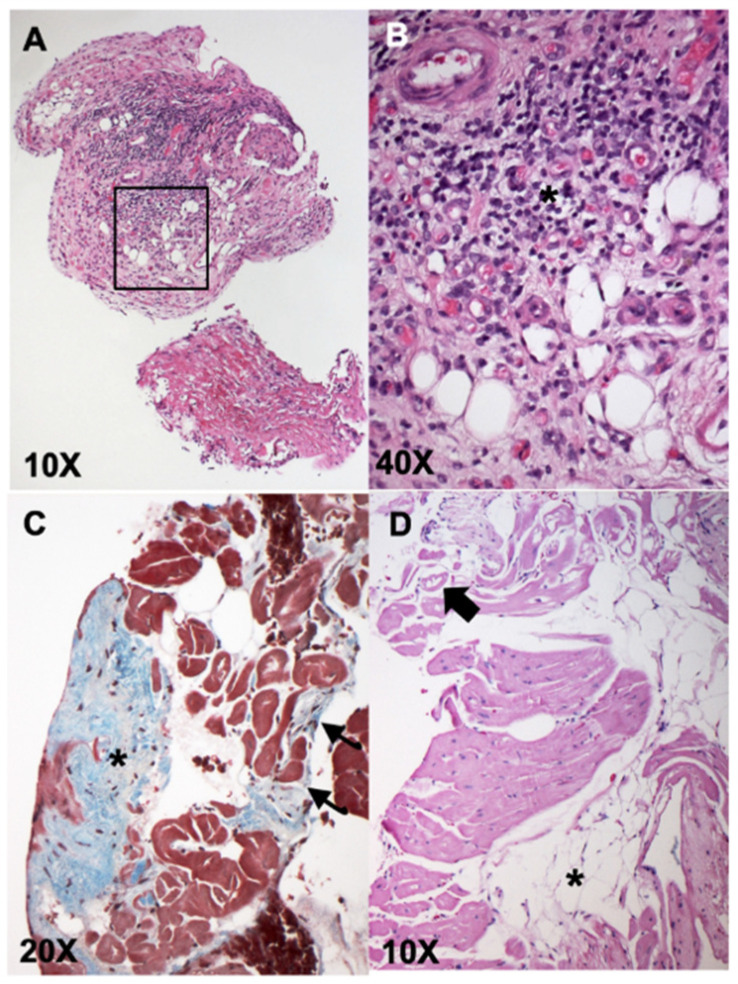
Representative histopathologic images from endomyocardial biopsies (EMBs) showing lymphoplasmacytic myocarditis (**A**,**B**), and arrhythmogenic right ventricular cardiomyopathy (ARVC) (**C**,**D**). Panels **A** (10×) and **B** (40×—from black box inset in (**A**)): H&E stained sample diagnostic for lymphoplasmacytic myocarditis (dog #29). These images show an endocardial lining with marked fibrotic thickening and massive lymphoplasmacellular infiltrate (purple infiltrate, asterisk, panel (**B**)); associated with fibrosis with neovascularization (vascular granulation tissue). Panel (**C**): Masson’s trichrome stain revealing focal subendocardial fibrosis (asterisk) and multifocal interstitial fibrosis (arrows). Panel (**D**) shows H&E stain revealing multifocal groups of adipocytes (asterisks) associated with fibrosis and rarefaction and vacuolization of the cardiomyocytes (arrow). The findings in Panels (**C**,**D**) contributed to the diagnosis of ARVC in dog #49.

**Figure 2 vetsci-08-00248-f002:**
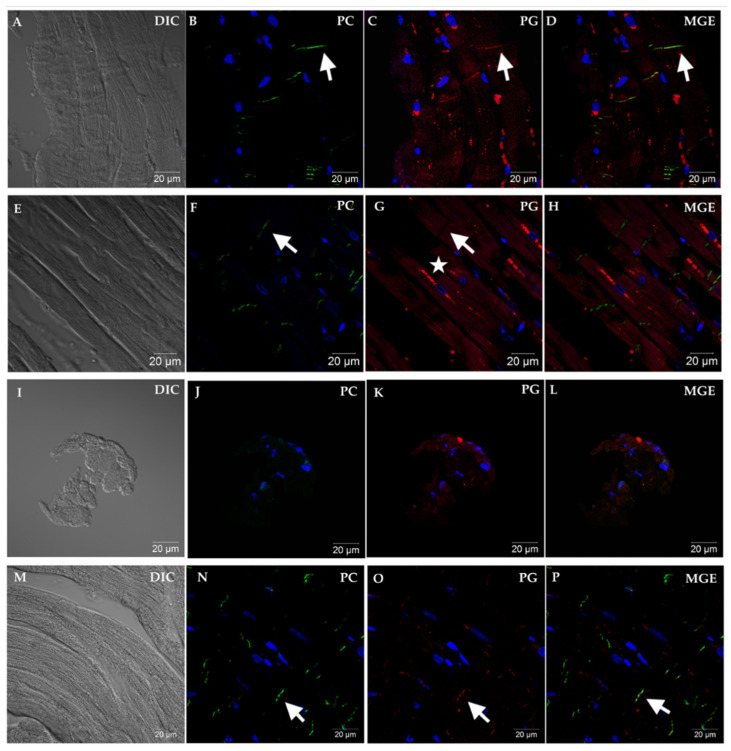
Representative immunofluorescent images showing PC and PG cellular localization from EMB sections of four different dogs. Panels (**A**–**D**) represent non-ARVC cardiac disease in a Dogue de Bordeaux (Dog #25) diagnosed with viral myocarditis. Note the longitudinal orientation of fibers (**A**), and positive immunofluorescent signal for PC at the area representing the ID (**B**). Immunofluorescent signal for PG (**C**) is also localized to the area of the ID and merges with PC signal (**D**). Panels (**E**–**H**) represent advanced (clinical) ARVC in an English bulldog (Dog #43). Note that while PC remains localized to the area of the ID (arrow, (**F**)), a complete absence of PG is noted in this area (arrow, (**G**)). Instead, PG is localized in a perinuclear pattern (star) and does not colocalize with PC (**H**). Panels (**I**–**L**) represent a non-diagnostic sample from a Great Dane (Dog #17) with tachycardia-induced cardiomyopathy. Panel I reveals the small number and imperfect orientation of cells, and lack of obvious cardiomyocytes, as evidenced in the non-diagnostic samples. Absence of PC signal was criteria for non-diagnostic sample (**J**). Panels (**M**–**P**) show representative images from a non-clinical boxer (Dog #44) with moderate ventricular ectopic load. Merged (MGE, (**D**,**H**,**L**,**P**)) shows colocalization, when present, of PC with PG (arrows). Images were obtained at 63× magnification.

## Data Availability

Not applicable.

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
