# Peer review of "Immunofluorescent Localization of Plakoglobin Is Altered in Endomyocardial Biopsy Samples from Dogs with Clinically Relevant Arrhythmogenic Right Ventricular Cardiomyopathy (ARVC)"

_vetsci, 2021, doi:10.3390/vetsci8110248_

Round 1
Reviewer 1 Report
Overall comments:
First of all, I congratulate the authors for their paper. I realize the great effort to perform studies to increase the knowledge of a particular disease.
The article is quite original, so I think it is very interesting.
However, sometimes it is somewhat confusing.
As a general comment, I recommend do not repeat the definition of the abbreviations along the text.
Specific comments:
Line 34-45: I recommend unifying the way to define the abbreviation. Sometimes the abbreviations precedes the definitions, others the definitions precedes the abbreviation. Please unify the rule.
Line 85-90: The author described 42 client-owned dogs plus 6 clinically healthy boxer used for the second part of the study. This is equal to 48 dogs. However in supplementary table 1 there are data from 49 dogs. In fact, in the abstract and others parts of the text, the author referred to 49 animals. Please clarify.
Line 96: clarify what is :”mild to moderate ventricular ectopic load” or included a cited to understand this classification.
Line 98: The abbreviation ARVC has been defined previously in line 47, I don’t think this is necessary again.
The same is applicable to line 106 with RV, line 162 with EMBs, line 164 with CU, line 166 with VA, line 192 with GDV, line 205 with PG and ID, line 207 with PG-MIS, line 209 PG-ID, line 233 with PG and EMB.
Line 106: I assumed that ARVC occult means no clinical signs , please clarify.
Line 113: In this occasion the authors do not use the abbreviation for Cornell University, previously defined.
Line 147-148: the abbreviations PG-ID and PG-MIS have not been defined.
Line 150: the reference 33 appear in the text, before the refence number 22
Line 305: abbreviation PG-IF is not defined, I think it could be a mistake.
Conclusion:
In my opinion the article needs some light corrections.
Author Response
We thank the reviewer for the detailed review and positive remarks regarding our manuscript. We have addressed the concerns.
Overall comments:
First of all, I congratulate the authors for their paper. I realize the great effort to perform studies to increase the knowledge of a particular disease.
The article is quite original, so I think it is very interesting.
However, sometimes it is somewhat confusing.
As a general comment, I recommend do not repeat the definition of the abbreviations along the text.
Specific comments:
Line 34-45: I recommend unifying the way to define the abbreviation. Sometimes the abbreviations precedes the definitions, others the definitions precedes the abbreviation. Please unify the rule.
Thank you for noting this error. It has been corrected.
Line 85-90: The author described 42 client-owned dogs plus 6 clinically healthy boxer used for the second part of the study. This is equal to 48 dogs. However in supplementary table 1 there are data from 49 dogs. In fact, in the abstract and others parts of the text, the author referred to 49 animals. Please clarify.
We apologize for the confusion and are grateful that the reviewer detected this clerical error. We have corrected Line 91 to read “43 client-owned dogs”
Line 96: clarify what is :”mild to moderate ventricular ectopic load” or included a cited to understand this classification.
Line 102-103: We have defined mild to moderate VE here, and referred the reader to the Supplemental Table.
Line 98: The abbreviation ARVC has been defined previously in line 47, I don’t think this is necessary again.
Corrected.
The same is applicable to line 106 with RV, line 162 with EMBs, line 164 with CU, line 166 with VA, line 192 with GDV, line 205 with PG and ID, line 207 with PG-MIS, line 209 PG-ID, line 233 with PG and EMB.
Corrected
Line 106: I assumed that ARVC occult means no clinical signs , please clarify.
Clarified in the text.
Line 113: In this occasion the authors do not use the abbreviation for Cornell University, previously defined.
Corrected
Line 147-148: the abbreviations PG-ID and PG-MIS have not been defined.
Lines 158-160: abbreviations have been better defined.
Line 150: the reference 33 appear in the text, before the refence number 22
We thank the reviewer for noting this clerical error. References have been corrected in the text.
Line 305: abbreviation PG-IF is not defined, I think it could be a mistake.
This has been corrected in the text.
Conclusion:
In my opinion the article needs some light corrections.
Thank you for your detailed review. We have made all corrections listed.

Reviewer 2 Report
In this study, Oxford et al., have the goal to demonstrate that a reduction in plakoglobin (PG) signal at intercalated disc in endomyocardial biopsy samples (EMBs) from dogs corresponds with a histopathologic diagnosis of arrhythmogenic right ventricular cardiomyopathy (ARVC). The authors concluded that the PG is reduced in dog with advanced form of ARVC, but not in the first phases when the pathology is hidden.
In my opinion, the manuscript is simple but well written, but I suggest to better organizing the table S1 for greater clarity of the data. For example:
-
In the table should be indicated the group to which each dog belongs (non-ARVC cardiac disease, clinical ARVC, and occult ARVC);
-
The table should be report clearly in which dog histopathology analysis highlighted the presence of ARVC characteristics;
-
The table should be report the classification described in 2.4 paragraph of material and methods section for the confocal microscopy analysis (PG-ID and PG-MIS).
Moreover, writing a caption for the table could help to clarify all this points.
Other minor suggestions:
-
Abbreviations have been reported twice in the text (after the abstract and at the end of discussion).
-
There are some superscript letters in the text (line 110 “bioptomeg”, line 133 “anti-rabbitk”, line 130 dyej), they have a particular meaning or are typos?
-
Line 226-230 should be reported in the material and methods section.
-
Line 108 correct the sentences “a total of three to six 3mm3”.
Author Response
Reviewer 2:
In this study, Oxford et al., have the goal to demonstrate that a reduction in plakoglobin (PG) signal at intercalated disc in endomyocardial biopsy samples (EMBs) from dogs corresponds with a histopathologic diagnosis of arrhythmogenic right ventricular cardiomyopathy (ARVC). The authors concluded that the PG is reduced in dog with advanced form of ARVC, but not in the first phases when the pathology is hidden.
We thank the reviewer for the detailed review and positive remarks regarding our manuscript. We have addressed the concerns.
In my opinion, the manuscript is simple but well written, but I suggest to better organizing the table S1 for greater clarity of the data. For example:
- In the table should be indicated the group to which each dog belongs (non-ARVC cardiac disease, clinical ARVC, and occult ARVC);
- The table should be report clearly in which dog histopathology analysis highlighted the presence of ARVC characteristics;
- The table should be report the classification described in 2.4 paragraph of material and methods section for the confocal microscopy analysis (PG-ID and PG-MIS).
Moreover, writing a caption for the table could help to clarify all this points.
The authors apologize that the supplemental table caption was not included in the submitted manuscript. The supplemental table is greatly improved with the reviewer’s suggestions incorporated, and the supplemental table legend is now included. Both the table and legend are found after the references at the end of the edited manuscript.
Other minor suggestions:
- Abbreviations have been reported twice in the text (after the abstract and at the end of discussion).
The abbreviations at the end of the discussion have been deleted.
- There are some superscript letters in the text (line 110 “bioptomeg”, line 133 “anti-rabbitk”, line 130 dyej), they have a particular meaning or are typos?
These superscripts have been corrected to correspond with the product information listed Lines 356-362.
- Line 226-230 should be reported in the material and methods section.
We have moved this information to Lines 356-362, to correspond with the superscript, referred to above. We will gladly follow the editorial guidelines for how to report product information.
- Line 108 correct the sentences “a total of three to six 3mm3”.
Line 115-116: this sentence has been edited for clarity.

Reviewer 3 Report
The manuscript is very interesting and I think it can be published in its current form
Author Response
We thank the reviewer for their kind words and thoughtful review.